# Flexible sampling of discrete data correlations without the marginal distributions

**Alfredo Kalaitzis**
Department of Statistical Science and CSML
University College London
a.kalaitzis@ucl.ac.uk

**Ricardo Silva**
Department of Statistical Science and CSML
University College London
ricardo@stats.ucl.ac.uk

## Abstract

Learning the joint dependence of discrete variables is a fundamental problem in machine learning, with many applications including prediction, clustering and dimensionality reduction. More recently, the framework of copula modeling has gained popularity due to its modular parameterization of joint distributions. Among other properties, copulas provide a recipe for combining flexible models for univariate marginal distributions with parametric families suitable for potentially high dimensional dependence structures. More radically, the extended rank likelihood approach of Hoff (2007) bypasses learning marginal models completely when such information is ancillary to the learning task at hand as in, e.g., standard dimensionality reduction problems or copula parameter estimation. The main idea is to represent data by their observable rank statistics, ignoring any other information from the marginals. Inference is typically done in a Bayesian framework with Gaussian copulas, and it is complicated by the fact this implies sampling within a space where the number of constraints increases quadratically with the number of data points. The result is slow mixing when using off-the-shelf Gibbs sampling. We present an efficient algorithm based on recent advances on constrained Hamiltonian Markov chain Monte Carlo that is simple to implement and does not require paying for a quadratic cost in sample size.

## 1 Contribution

There are many ways of constructing multivariate discrete distributions: from full contingency tables in the small dimensional case [1], to structured models given by sparsity constraints [11] and (hierarchies of) latent variable models [6]. More recently, the idea of *copula modeling* [16] has been combined with such standard building blocks. Our contribution is a novel algorithm for efficient Markov chain Monte Carlo (MCMC) for the copula framework introduced by [7], extending algorithmic ideas introduced by [17].

A copula is a continuous cumulative distribution function (CDF) with uniformly distributed univariate marginals in the unit interval $[0, 1]$. It complements graphical models and other formalisms that provide a modular parameterization of joint distributions. The core idea is simple and given by the following observation: suppose we are given a (say) bivariate CDF $F(y_1, y_2)$ with marginals $F_1(y_1)$ and $F_2(y_2)$. This CDF can then be rewritten as $F(F_1^{-1}(F_1(y_1)), F_2^{-1}(F_2(y_2)))$. The function $C(\cdot, \cdot)$ given by $F(F_1^{-1}(\cdot), F_2^{-1}(\cdot))$ is a copula. For discrete distributions, this decomposition is not unique but still well-defined [16]. Copulas have found numerous applications in statistics and machine learning since they provide a way of constructing flexible multivariate distributions by mix-and-matching different copulas with different univariate marginals. For instance, one can combine flexible univariate marginals $F_i(\cdot)$ with useful but more constrained high-dimensional copulas. We will not further motivate the use of copula models, which has been discussed at length in recent

machine learning publications and conference workshops, and for which comprehensive textbooks exist [e.g., 9]. For a recent discussion on the applications of copulas from a machine learning perspective, [4] provides an overview. [10] is an early reference in machine learning. The core idea dates back at least to the 1950s [16].

In the discrete case, copulas can be difficult to apply: transforming a copula CDF into a probability mass function (PMF) is computationally intractable in general. For the continuous case, a common trick goes as follows: transform variables by defining $a_i \equiv \hat{F}_i(y_i)$ for an estimate of $F_i(\cdot)$ and then fit a copula density $c(\cdot, \ldots, \cdot)$ to the resulting $a_i$ [e.g. 9]. It is not hard to check this breaks down in the discrete case [7]. An alternative is to represent the CDF to PMF transformation for each data point by a continuous integral on a bounded space. Sampling methods can then be used. This trick has allowed many applications of the Gaussian copula to discrete domains. Readers familiar with probit models will recognize the similarities to models where an underlying latent Gaussian field is discretized into observable integers as in Gaussian process classifiers and ordinal regression [18]. Such models can be indirectly interpreted as special cases of the Gaussian copula.

In what follows, we describe in Section 2 the Gaussian copula and the general framework for constructing Bayesian estimators of Gaussian copulas by [7], the extended rank likelihood framework. This framework entails computational issues which are discussed. A recent general approach for MCMC in constrained Gaussian fields by [17] can in principle be directly applied to this problem as a blackbox, but at a cost that scales quadratically in sample size and as such it is not practical in general. Our key contribution is given in Section 4. An application experiment on the Bayesian Gaussian copula factor model is performed in Section 5. Conclusions are discussed in the final section.

## 2   Gaussian copulas and the *extended rank likelihood*

It is not hard to see that any multivariate Gaussian copula is fully defined by a correlation matrix $\mathbf{C}$, since marginal distributions have no free parameters. In practice, the following equivalent generative model is used to define a sample $\mathbf{U}$ according to a Gaussian copula $\mathcal{GC}(\mathbf{C})$:

1. Sample $\mathbf{Z}$ from a zero mean Gaussian with covariance matrix $\mathbf{C}$
2. For each $Z_j$, set $U_j = \Phi(z_j)$, where $\Phi(\cdot)$ is the CDF of the standard Gaussian

It is clear that each $U_j$ follows a uniform distribution in $[0, 1]$. To obtain a model for variables $\{y_1, y_2, \ldots, y_p\}$ with marginal distributions $F_j(\cdot)$ and copula $\mathcal{GC}(\mathbf{C})$, one can add the deterministic step $y_j = F_j^{-1}(u_j)$. Now, given $n$ samples of observed data $\mathbf{Y} \equiv \{y_1^{(1)}, \ldots, y_p^{(1)}, y_1^{(2)}, \ldots, y_p^{(n)}\}$, one is interested on inferring $\mathbf{C}$ via a Bayesian approach and the posterior distribution

$$p(\mathbf{C}, \theta_F \mid \mathbf{Y}) \propto p_{\mathcal{GC}}(\mathbf{Y} \mid \mathbf{C}, \theta_F)\pi(\mathbf{C}, \theta_F)$$

where $\pi(\cdot)$ is a prior distribution, $\theta_F$ are marginal parameters for each $F_j(\cdot)$, which in general might need to be marginalized since they will be unknown, and $p_{\mathcal{GC}}(\cdot)$ is the PMF of a (here discrete) distribution with a Gaussian copula and marginals given by $\theta_F$.

Let $\mathbf{Z}$ be the underlying latent Gaussians of the corresponding copula for dataset $\mathbf{Y}$. Although $\mathbf{Y}$ is a deterministic function of $\mathbf{Z}$, this mapping is not invertible due to the discreteness of the distribution: each marginal $F_j(\cdot)$ has jumps. Instead, the reverse mapping only enforces the constraints where $y_j^{(i_1)} < y_j^{(i_2)}$ implies $z_j^{(i_1)} < z_j^{(i_2)}$. Based on this observation, [7] considers the event $\mathbf{Z} \in D(\mathbf{y})$, where $D(\mathbf{y})$ is the set of values of $\mathbf{Z}$ in $\mathbb{R}^{n \times p}$ obeying those constraints, that is

$$D(\mathbf{y}) \equiv \left\{ \mathbf{Z} \in \mathbb{R}^{n \times p} : \max \left\{ z_j^{(k)} s.t. \ y_j^{(k)} < y_j^{(i)} \right\} < z_j^{(i)} < \min \left\{ z_j^{(k)} s.t. \ y_j^{(i)} < y_j^{(k)} \right\} \right\}.$$

Since $\{\mathbf{Y} = \mathbf{y}\} \Rightarrow \mathbf{Z}(\mathbf{y}) \in D(\mathbf{y})$, we have

$$
\begin{aligned}
p_{\mathcal{GC}}(\mathbf{Y} \mid \mathbf{C}, \theta_F) &= p_{\mathcal{GC}}(\mathbf{Z} \in D(\mathbf{y}), \mathbf{Y} \mid \mathbf{C}, \theta_F) \\
&= p_{\mathcal{N}}(\mathbf{Z} \in D(\mathbf{y}) \mid \mathbf{C}) \times p_{\mathcal{GC}}(\mathbf{Y} \mid \mathbf{Z} \in D(\mathbf{y}), \mathbf{C}, \theta_F),
\end{aligned}
\tag{1}
$$

the first factor of the last line being that of a zero-mean a Gaussian density function marginalized over $D(\mathbf{y})$.

The extended rank likelihood is defined by the first factor of (1). With this likelihood, inference for $\mathbf{C}$ is given simply by marginalizing

$$p(\mathbf{C}, \mathbf{Z} \mid \mathbf{Y}) \propto I(\mathbf{Z} \in D(\mathbf{y})) \, p_{\mathcal{N}}(\mathbf{Z} \mid \mathbf{C}) \, \pi(\mathbf{C}), \qquad (2)$$

the first factor of the right-hand side being the usual binary indicator function.

Strictly speaking, this is not a fully Bayesian method since partial information on the marginals is ignored. Nevertheless, it is possible to show that under some mild conditions there is information in the extended rank likelihood to consistently estimate $\mathbf{C}$ [13]. It has two important properties: first, in many applications where marginal distributions are nuisance parameters, this sidesteps any major assumptions about the shape of $\{F_i(\cdot)\}$ – applications include learning the degree of dependence among variables (e.g., to understand relationships between social indicators as in [7] and [13]) and copula-based dimensionality reduction (a generalization of correlation-based principal component analysis, e.g., [5]); second, MCMC inference in the extended rank likelihood is conceptually simpler than with the joint likelihood, since dropping marginal models will remove complicated entanglements between $\mathbf{C}$ and $\theta_F$. Therefore, even if $\theta_F$ is necessary (when, for instance, predicting missing values of $\mathbf{Y}$), an estimate of $\mathbf{C}$ can be computed separately and will not depend on the choice of estimator for $\{F_i(\cdot)\}$. The standard model with a full correlation matrix $\mathbf{C}$ can be further refined to take into account structure implied by sparse inverse correlation matrices [2] or low rank decompositions via higher-order latent variable models [13], among others. We explore the latter case in section 5.

An off-the-shelf algorithm for sampling from (2) is full Gibbs sampling: first, given $\mathbf{Z}$, the (full or structured) correlation matrix $\mathbf{C}$ can be sampled by standard methods. More to the point, sampling $\mathbf{Z}$ is straightforward if for each variable $j$ and data point $i$ we sample $Z_j^{(i)}$ conditioned on all other variables. The corresponding distribution is an univariate truncated Gaussian. This is the approach used originally by Hoff. However, mixing can be severely compromised by the sampling of $\mathbf{Z}$, and that is where novel sampling methods can facilitate inference.

## 3   Exact HMC for truncated Gaussian distributions

Hoff's algorithm modifies the positions of all $Z_j^{(i)}$ associated with a particular discrete value of $Y_j$, conditioned on the remaining points. As the number of data points increases, the spread of the hard boundaries on $Z_j^{(i)}$, given by data points of $Z_j$ associated with other levels of $Y_j$, increases. This reduces the space in which variables $Z_j^{(i)}$ can move at a time.

To improve the mixing, we aim to sample from the *joint* Gaussian distribution of all latent variables $Z_j^{(i)}$, $i = 1 \ldots n$, conditioned on other columns of the data, such that the constraints between them are satisfied and thus the ordering in the observation level is conserved. Standard Gibbs approaches for sampling from truncated Gaussians reduce the problem to sampling from univariate truncated Gaussians. Even though each step is computationally simple, mixing can be slow when strong correlations are induced by very tight truncation bounds.

In the following, we briefly describe the methodology recently introduced by [17] that deals with the problem of sampling from $\log p(\mathbf{x}) \propto -\frac{1}{2}\mathbf{x}^\top \mathbf{M} \mathbf{x} + \mathbf{r}^\top \mathbf{x}$, where $\mathbf{x}, \mathbf{r} \in \mathbb{R}^n$ and $\mathbf{M}$ is positive definite, with linear constraints of the form $\mathbf{f}_j^\top \mathbf{x} \leq g_j$, where $\mathbf{f}_j \in \mathbb{R}^n$, $j = 1 \ldots m$, is the normal vector to some linear boundary in the sample space.

Later in this section we shall describe how this framework can be applied to the Gaussian copula extended rank likelihood model. More importantly, the observed rank statistics impose only linear constraints of the form $x_{i_1} \leq x_{i_2}$. We shall describe how this special structure can be exploited to reduce the runtime complexity of the constrained sampler from $\mathcal{O}(n^2)$ (in the number of observations) to $\mathcal{O}(n)$ in practice.

### 3.1   Hamiltonian Monte Carlo for the Gaussian distribution

Hamiltonian Monte Carlo (HMC) [15] is a MCMC method that extends the sampling space with auxiliary variables so that (ideally) deterministic moves in the joint space brings the sampler to

potentially far places in the original variable space. Deterministic moves cannot in general be done, but this is possible in the Gaussian case.

The form of the Hamiltonian for the general $d$-dimensional Gaussian case with mean $\boldsymbol{\mu}$ and precision matrix $\mathbf{M}$ is:

$$H = \frac{1}{2}\,\mathbf{x}^\top \mathbf{M}\mathbf{x} - \mathbf{r}^\top \mathbf{x} + \frac{1}{2}\,\mathbf{s}^\top \mathbf{M}^{-1}\mathbf{s}\,, \tag{3}$$

where $\mathbf{M}$ is also known in the present context as the *mass* matrix, $\mathbf{r} = \mathbf{M}\boldsymbol{\mu}$ and $\mathbf{s}$ is the *velocity*. Both $\mathbf{x}$ and $\mathbf{s}$ are Gaussian distributed so this Hamiltonian can be seen (up to a constant) as the negative log of the product of two independent Gaussian random variables. The physical interpretation is that of a sum of *potential* and *kinetic* energy terms, where the *total* energy of the system is conserved.

In a system where this Hamiltonian function is constant, we can exactly compute its evolution through the pair of differential equations:

$$\dot{\mathbf{x}} = \nabla_s H = \mathbf{M}^{-1}\mathbf{s}\,, \qquad \dot{\mathbf{s}} = -\nabla_x H = -\mathbf{M}\mathbf{x} + \mathbf{r}\,. \tag{4}$$

These are solved exactly by $\quad \mathbf{x}(t) = \boldsymbol{\mu} + \mathbf{a}\sin(t) + \mathbf{b}\cos(t) \quad$, where $\mathbf{a}$ and $\mathbf{b}$ can be identified at initial conditions $(t = 0)$:

$$\mathbf{a} = \dot{\mathbf{x}}(0) = \mathbf{M}^{-1}s\,, \qquad \mathbf{b} = \mathbf{x}(0) - \boldsymbol{\mu}\,. \tag{5}$$

Therefore, the exact HMC algorithm can be summarised as follows:

- Initialise the allowed travel time $T$ and some initial position $\mathbf{x}_0$.
- Repeat for HMC samples $k = 1 \ldots N$
  1. Sample $\mathbf{s}_k \sim \mathcal{N}(\mathbf{0}, \mathbf{M})$
  2. Use $\mathbf{s}_k$ and $\mathbf{x}_k$ to update $\mathbf{a}$ and $\mathbf{b}$ and store the new position at the end of the trajectory $\mathbf{x}_{k+1} = \mathbf{x}(T)$ as an HMC sample.

It can be easily shown that the Markov chain of sampled positions has the desired equilibrium distribution $\mathcal{N}(\boldsymbol{\mu}, \mathbf{M}^{-1})$ [17].

### 3.2 Sampling with linear constraints

Sampling from multivariate Gaussians does not require any method as sophisticated as HMC, but the plot thickens when the target distribution is truncated by linear constraints of the form $\mathbf{Fx} \le \mathbf{g}$. Here, $\mathbf{F} \in \mathbb{R}^{m \times n}$ is a constraint matrix whose every row is the normal vector to a linear boundary in the sample space. This is equivalent to sampling from a Gaussian that is confined in the (not necessarily bounded) convex polyhedron $\{\mathbf{x} : \mathbf{Fx} \le \mathbf{g}\}$. In general, to remain within the boundaries of each wall, once a new velocity has been sampled one must compute all possible collision times with the walls. The smallest of all collision times signifies the wall that the particle should *bounce* from at that collision time. Figure 1 illustrates the concept with two simple examples on 2 and 3 dimensions.

The collision times can be computed analytically and their equations can be found in the supplementary material. We also point the reader to [17] for a more detailed discussion of this implementation. Once the wall to be hit has been found, then position and velocity at impact time are computed and the velocity is reflected about the boundary normal[1]. The constrained HMC sampler is summarized follows:

- Initialise the allowed travel time $T$ and some initial position $\mathbf{x}_0$.
- Repeat for HMC samples $k = 1 \ldots N$
  1. Sample $\mathbf{s}_k \sim \mathcal{N}(\mathbf{0}, \mathbf{M})$
  2. Use $\mathbf{s}_k$ and $\mathbf{x}_k$ to update $\mathbf{a}$ and $\mathbf{b}$.

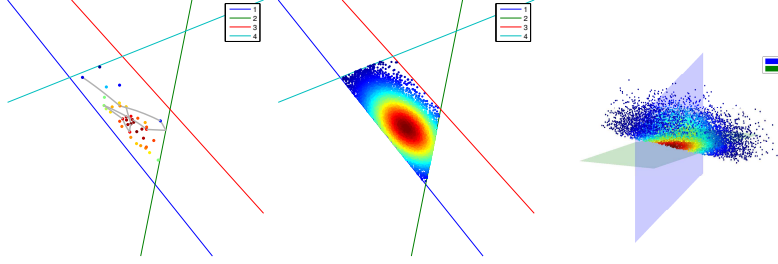

Figure 1: **Left**: Trajectories of the first 40 iterations of the exact HMC sampler on a 2D truncated Gaussian. A reflection of the velocity can clearly be seen when the particle meets wall #2 . Here, the constraint matrix $\mathbf{F}$ is a $4 \times 2$ matrix. **Center**: The same example after 40000 samples. The coloring of each sample indicates its density value. **Right**: The anatomy of a 3D Gaussian. The walls are now planes and in this case $\mathbf{F}$ is a $2 \times 3$ matrix. Figure best seen in color.

3. Reset remaining travel time $T_{\text{left}} \leftarrow T$ . Until no travel time is left or no walls can be reached (no solutions exist), do:
   (a) Compute impact times with all walls and pick the smallest one, $t_h$ (if a solution exists).
   (b) Compute $\mathbf{v}(t_h)$ and reflect it about the hyperplane $\mathbf{f}_h$ . This is the updated velocity after impact. The updated position is $\mathbf{x}(t_h)$ .
   (c) $T_{\text{left}} \leftarrow T_{\text{left}} - t_h$
4. Store the new position at the end of the trajectory $\mathbf{x}_{k+1}$ as an HMC sample.

In general, all walls are candidates for impact, so the runtime of the sampler is linear in $m$ , the number of constraints. This means that the computational load is concentrated in step 3(a). Another consideration is that of the *allocated travel time* $T$ . Depending on the shape of the bounding polyhedron and the number of walls, a very large travel time can induce many more bounces thus requiring more computations per sample. On the other hand, a very small travel time explores the distribution more locally so the mixing of the chain can suffer. What constitutes a given travel time "large" or "small" is relative to the dimensionality, the number of constraints and the structure of the constraints.

Due to the nature of our problem, the number of constraints, when explicitly expressed as linear functions, is $\mathcal{O}(n^2)$ . Clearly, this restricts any direct application of the HMC framework for Gaussian copula estimation to small-sample $(n)$ datasets. More importantly, we show how to exploit the structure of the constraints to reduce the number of candidate walls (prior to each bounce) to $\mathcal{O}(n)$ .

## 4  HMC for the Gaussian Copula extended rank likelihood model

Given some discrete data $\mathbf{Y} \in \mathbb{R}^{n \times p}$ , the task is to infer the correlation matrix of the underlying Gaussian copula. Hoff's sampling algorithm proceeds by alternating between sampling the *continuous latent* representation $Z_j^{(i)}$ of each $Y_j^{(i)}$, for $i = 1 \ldots n$, $j = 1 \ldots p$, and sampling a covariance matrix from an inverse-Wishart distribution conditioned on the sampled matrix $\mathbf{Z} \in \mathbb{R}^{n \times p}$ , which is then renormalized as a correlation matrix.

From here on, we use matrix notation for the samples, as opposed to the random variables – with $Z_{i,j}$ replacing $Z_j^{(i)}$, $\mathbf{Z}_{:,j}$ being a column of $\mathbf{Z}$, and $\mathbf{Z}_{:,\backslash j}$ being the submatrix of $\mathbf{Z}$ without the $j$-th column.

In a similar vein to Hoff's sampling algorithm, we replace the successive sampling of each $Z_{i,j}$ conditioned on $\mathbf{Z}_{i,\backslash j}$ (a conditional univariate truncated Gaussian) with the simultaneous sampling of $\mathbf{Z}_{:,j}$ conditioned on $\mathbf{Z}_{:,\backslash j}$. This is done through an HMC step from a conditional multivariate truncated Gaussian.

The added *benefit* of this HMC step over the standard Gibbs approach, is that of a *handle* for regulating the *trade-off* between *exploration* and *runtime* via the allocated travel time $T$. Larger travel times potentially allow for larger moves in the sample space, but it comes at a cost as explained in the sequel.

### 4.1 The *Hough envelope* algorithm

**The special structure of constraints.** Recall that the number of constraints is quadratic in the dimension of the distribution. This is because every $\mathbf{Z}$ sample must satisfy the conditions of the event $\mathbf{Z} \in D(\mathbf{y})$ of the extended rank likelihood (see Section 2). In other words, for any column $Z_{:,j}$, all entries are organised into a partition $L^{(j)}$ of $|L^{(j)}|$ levels, the number of unique values observed for the discrete or ordinal variable $Y^{(j)}$. Thereby, for any two adjacent levels $l_k, l_{k+1} \in L^{(j)}$ and any pair $i_1 \in l_k$, $i_2 \in l_{k+1}$, it must be true that $Z_{l_i,j} < Z_{l_{i+1},j}$. Equivalently, a constraint $\mathbf{f}$ exists where $f_{i_1} = 1$, $f_{i_2} = -1$ and $g = 0$. It is easy to see that $\mathcal{O}(n^2)$ of such constraints are induced by the order statistics of the $j$-th variable. To deal with this boundary explosion, we developed the *Hough Envelope* algorithm to search efficiently, within all pairs in $\{\mathbf{Z}_{:,j}\}$, in practically *linear* time.

Recall in HMC (section 3.2) that the trajectory of the particle, $\mathbf{x}(t)$, is decomposed as

$$x_i(t) = a_i \sin(t) + b_i \cos(t) + \mu_i , \tag{6}$$

and there are $n$ such functions, grouped into a partition of levels as described above. The Hough envelope[2] is found for every pair of adjacent levels. We illustrate this with an example of 10 dimensions and two levels in Figure 2, without loss of generalization to any number of levels or dimensions. Assume we represent trajectories for points in level $l_k$ with blue curves, and points in $l_{k+1}$ with red curves. Assuming we start with a valid state, at time $t = 0$ all red curves are above all blue curves. The goal is to find the smallest $t$ where a blue curve meets a red curve. This will be our collision time where a bounce will be necessary.

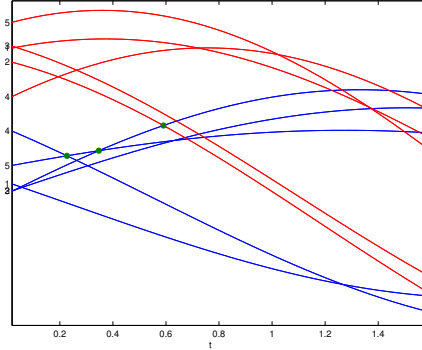

Figure 2: The trajectories $x_j(t)$ of each component are sinusoid functions. The right-most green dot signifies the wall and the time $t_h$ of the earliest bounce, where the first inter-level pair (that is, any two components respectively from the blue and red level) becomes equal, in this case the constraint *activated* being $x_{blue_2} = x_{red_2}$.

1. First we find the largest component $blue_{max}$ of the blue level at $t = 0$. This takes $\mathcal{O}(n)$ time. Clearly, this will be the largest component until its sinusoid intersects that of any other component.

2. To find the next largest component, compute the roots of $x_{blue_{max}}(t) - x_i(t) = 0$ for all components and pick the smallest (earliest) one (represented by a green dot). This also takes $\mathcal{O}(n)$ time.

3. Repeat this procedure until a red sinusoid crosses the highest running blue sinusoid. When this happens, the time of earliest bounce and its constraint are found.

In the worst-case scenario, $n$ such repetitions have to be made, but in practice we can safely assume an fixed upper bound $h$ on the number of blue crossings before a inter-level crossing occurs. In experiments, we found $h << n$, no more than 10 in simulations with hundreds of thousands of curves. Thus, this search strategy takes $\mathcal{O}(n)$ time in practice to complete, mirroring the analysis of other output-sensitive algorithms such as the gift wrapping algorithm for computing convex hulls [8]. Our HMC sampling approach is summarized in **Algorithm 1**.

**Algorithm 1** HMC for GCERL

---

\# Notation: $\mathcal{TMN}(\boldsymbol{\mu}, \mathbf{C}, \mathbf{F})$ is a truncated multivariate normal with location vector $\boldsymbol{\mu}$, scale matrix $\mathbf{C}$ and constraints encoded by $\mathbf{F}$ and $\mathbf{g} = \mathbf{0}$.

\# $\mathcal{IW}(df, \mathbf{V}_0)$ is an *inverse-Wishart* prior with degrees of freedom $df$ and scale matrix $\mathbf{V}_0$.

**Input:** $\mathbf{Y} \in \mathbb{R}^{n \times p}$, allocated travel time $T$, a starting $\mathbf{Z}$ and variable covariance $\mathbf{V} \in \mathbb{R}^{p \times p}$, $df = p + 2$, $\mathbf{V}_0 = df\mathbf{I}_p$ and chain size $N$.

Generate constraints $\mathbf{F}^{(j)}$ from $\mathbf{Y}_{:,j}$, for $j = 1 \ldots p$.

**for** samples $k = 1 \ldots N$ **do**

    \# Resample $\mathbf{Z}$ as follows:

    **for** variables $j = 1 \ldots p$ **do**

        Compute parameters: $\sigma_j^2 = \mathbf{V}_{jj} - \mathbf{V}_{j,\backslash j}\mathbf{V}_{\backslash j,\backslash j}^{-1}\mathbf{V}_{\backslash j,j}$,      $\boldsymbol{\mu}_j = \mathbf{Z}_{:,\backslash j}\mathbf{V}_{\backslash j,\backslash j}^{-1}\mathbf{V}_{\backslash j,j}$.

        Get one sample $\mathbf{Z}_{:,j} \sim \mathcal{TMN}\left(\boldsymbol{\mu}_j, \sigma_j^2\mathbf{I}, \mathbf{F}^{(j)}\right)$ efficiently by using the *Hough Envelope* algorithm, see section 4.1.

    **end for**

    Resample $\mathbf{V} \sim \mathcal{IW}(df + n, \mathbf{V}_0 + \mathbf{Z}^\top\mathbf{Z})$.

    Compute correlation matrix $\mathbf{C}$, s.t. $C_{i,j} = \mathbf{V}_{i,j}/\sqrt{V_{i,i}V_{j,j}}$ and store sample, $\mathbf{C}^{(k)} \leftarrow \mathbf{C}$.

**end for**

---

## 5 An application on the Bayesian Gausian copula factor model

In this section we describe an experiment that highlights the benefits of our HMC treatment, compared to a state-of-the-art parameter expansion (PX) sampling scheme. During this experiment we ask the important question:

*"How do the two schemes compare when we exploit the full-advantage of the HMC machinery to jointly sample parameters and the augmented data Z, in a model of latent variables and structured correlations?"*

We argue that under such circumstances the superior convergence speed and mixing of HMC undeniably compensate for its computational overhead.

**Experimental setup**    In this section we provide results from an application on the Gaussian copula latent factor model of [13] (Hoff's model [7] for low-rank structured correlation matrices). We modify the parameter expansion (PX) algorithm used by [13] by replacing two of its Gibbs steps with a single HMC step. We show a much faster convergence to the true mode with considerable support on its vicinity. We show that unlike the HMC, the PX algorithm falls short of properly exploring the posterior in any reasonable finite amount of time, even for small models, even for small samples. Worse, *PX fails in ways one cannot easily detect*.

Namely, we sample each row of the factor loadings matrix $\boldsymbol{\Lambda}$ jointly with the corresponding column of the augmented data matrix $\mathbf{Z}$, conditioning on the higher-order latent factors. This step is analogous to Pakman and Paninski's [17, sec.3.1] use of HMC in the context of a binary probit model (the extension to many levels in the discrete marginal is straightforward with direct application of the constraint matrix $\mathbf{F}$ and the *Hough envelope* algorithm). The sampling of the higher level latent factors remains identical to [13]. Our scheme involves no parameter expansion. We do however *interweave* the Gibbs step for the $\mathbf{Z}$ matrix similarly to Hoff. This has the added benefit of exploring the $\mathbf{Z}$ sample space within their current boundaries, complementing the joint $(\boldsymbol{\lambda}, \mathbf{z})$ sampling which moves the boundaries jointly. The value of such "*interweaving*" schemes has been addressed in [19].

**Results**    We perform simulations of 10000 iterations, $n = 1000$ observations (rows of $\mathbf{Y}$), travel time $\pi/2$ for HMC with the setups listed in the following table, along with the elapsed times of each sampling scheme. These experiments were run on Intel COREi7 desktops with 4 cores and 8GB of RAM. Both methods were parallelized across the observed variables (p).

| Figure | p (vars) | k (latent factors) | M (ordinal levels) | elapsed (mins): HMC | PX |
|---|---|---|---|---|---|
| 3(a) : | 20 | 5 | 2 | 115 | 8 |
| 3(b) : | 10 | 3 | 2 | 80 | 6 |
| 3(c) : | 10 | 3 | 5 | 203 | 16 |

Many functionals of the loadings matrix $\boldsymbol{\Lambda}$ can be assessed. We focus on reconstructing the true (low-rank) correlation matrix of the Gaussian copula. In particular, we summarize the algorithm's

outcome with the **root mean squared error (RMSE)** of the differences between entries of the ground-truth correlation matrix and the implied correlation matrix at each iteration of a MCMC scheme (so the following plots looks like a time-series of 10000 timepoints), see Figures 3(a), 3(b) and 3(c) .

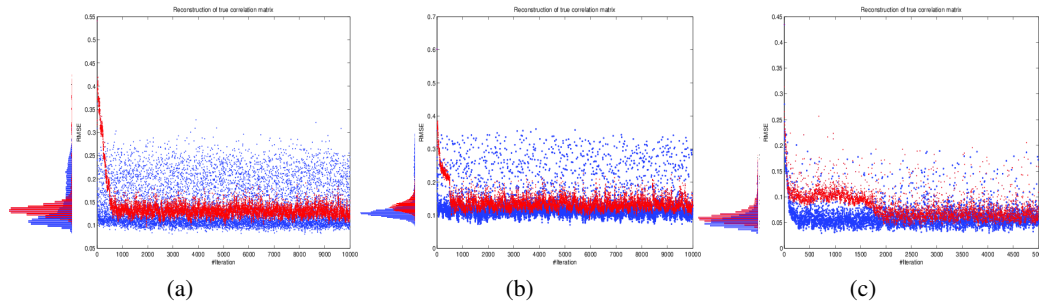

(a)&emsp;&emsp;&emsp;&emsp;&emsp;&emsp;(b)&emsp;&emsp;&emsp;&emsp;&emsp;&emsp;(c)

Figure 3: Reconstruction (RMSE per iteration) of the low-rank structured correlation matrix of the Gaussian copula and its histogram (along the left side).
**(a)** Simulation setup: 20 variables, 5 factors, 5 levels. HMC (blue) reaches a better mode faster (in *iterations/CPU-time*) than PX (red). Even more importantly the RMSE posterior samples of PX are concentrated in a much smaller region compared to HMC, even after 10000 iterations. This illustrates that PX poorly explores the true distribution.
**(b)** Simulation setup: 10 vars, 3 factors, 2 levels. We observe behaviors similar to Figure 3(a). Note that the histogram counts RMSEs after the burn-in period of PX (iteration #500).
**(c)** Simulation setup: 10 vars, 3 factors, 5 levels. We observe behaviors similar to Figures 3(a) and 3(b) but with a thinner tail for HMC. Note that the histogram counts RMSEs after the burn-in period of PX (iteration #2000).

**Main message**&emsp;&emsp;HMC reaches a better mode faster (iterations/CPUtime). Even more importantly the RMSE posterior samples of PX are concentrated in a much smaller region compared to HMC, even after 10000 iterations. This illustrates that PX poorly explores the true distribution. As an analogous situation we refer to the top and bottom panels of Figure 14 of Radford Neal's *slice sampler* paper [14]. If there was no comparison against HMC, there would be no evidence from the PX plot alone that the algorithm is performing poorly. This mirrors Radford Neal's statement opening Section 8 of his paper: *"a wrong answer is obtained without any obvious indication that something is amiss"*. The concentration on the posterior mode of PX in these simulations is misleading of the truth. PX might seen a bit simpler to implement, but it seems one cannot avoid using complex algorithms for complex models. We urge practitioners to revisit their past work with this model to find out by how much credible intervals of functionals of interest have been overconfident. Whether trivially or severely, our algorithm offers the first principled approach for checking this out.

# 6&emsp;Conclusion

Sampling large random vectors simultaneously in order to improve mixing is in general a very hard problem, and this is why clever methods such as HMC or elliptical slice sampling [12] are necessary. We expect that the method here developed is useful not only for those with data analysis problems within the large family of Gaussian copula extended rank likelihood models, but the method itself and its behaviour might provide some new insights on MCMC sampling in constrained spaces in general. Another direction of future work consists of exploring methods for elliptical copulas, and related possible extensions of general HMC for non-Gaussian copula models.

**Acknowledgements**

The quality of this work has benefited largely from comments by our anonymous reviewers and useful discussions with Simon Byrne and Vassilios Stathopoulos. Research was supported by EPSRC grant EP/J013293/1.

## Footnotes

[1]Also equivalent to transforming the velocity with a *Householder reflection matrix* about the bounding hyperplane.

[2]The name is inspired from the fact that each $x_i(t)$ is the sinusoid representation, in *angle-distance* space, of all lines that pass from the $(a_i, b_i)$ point in $a - b$ space. A representation known in image processing as the *Hough transform* [3].

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
