[Supplementary Material]

# Supplementary material: Flexible sampling of discrete data correlations without the marginal distributions

**Alfredo Kalaitzis**
Department of Statistical Science and CSML
University College London
a.kalaitzis@ucl.ac.uk

**Ricardo Silva**
Department of Statistical Science and CSML
University College London
ricardo@stats.ucl.ac.uk

## 1 Computing Collision Times Within HMC

More details of the following can be found in [Pakman, et al., 2012]. For each wall $j = 1 \ldots m$, a particle with velocity $\mathbf{a}$ and position $\mathbf{b}$ will hit when

$$
\begin{aligned}
K_j(t) &= \mathbf{f}_j^\top \mathbf{x} + g_j \\
&= \mathbf{f}_j^\top \mathbf{a} \sin(t) + \mathbf{f}_j^\top \mathbf{b} \cos(t) - g_j = 0 .
\end{aligned}
\tag{1}
$$

To find the root of $K_j(t) = 0$, we can express this through a scaled cosine with a phase term as follows:
Define

$$
A_j = \mathbf{f}_j^\top \mathbf{a}
\tag{2}
$$

$$
B_j = \mathbf{f}_j^\top \mathbf{b}
\tag{3}
$$

$$
u_j \equiv \sqrt{A_j^2 + B_j^2} \quad \text{and}
\tag{4}
$$

$$
\tan(\phi_j) \equiv -\frac{A_j}{B_j} .
\tag{5}
$$

Now (1) can be expressed as

$$
K_j(t) = u_j \left( \frac{A_j}{u_j} \sin(t) + \frac{B_j}{u_j} \cos(t) \right) - g_j
\tag{6}
$$

$$
= u_j \left( \widehat{A}_j \sin(t) + \widehat{B}_j \cos(t) \right) - g_j
\tag{7}
$$

$$
= u_j \left( -\sin(\phi_j) \sin(t) + \cos(\phi_j) \cos(t) \right) - g_j
\tag{8}
$$

$$
= u_j \cos(t + \phi_j) - g_j ,
\tag{9}
$$

which yields the solutions

$$
t = \pm \cos^{-1} \left( \frac{g_j}{u_j} \right) - \phi_j .
\tag{10}
$$

It can be shown that roots exist only for walls where $u_j > |g_j|$.

## 2 Full Correlation Matrix Plots

Figure 1: Summary of sampling behavior for the HMC (blue lines) and Hoff's algorithm (red lines) for the experiments in Section 5.

## 3 Illustration

In theory, one can only expect mixing improvements over the algorithm outlined by Hoff, and we do not believe the computational cost of HMC with rank likelihood constraints can be reduced by more than a constant factor in the worst case. Empirically, however, it is not obvious how these theoret-

Figure 2: Top row shows the evolution of particular copula correlation coefficients in our problem. The red line shows the evolution of samples taken by Hoff's algorithm, and blue is the corresponding one with the HMC component (truncated at 1000 samples for visualization purposes). Bottom row shows representative samples of the whole copula correlation matrix from the two chains at different stages (50th and 1000th).

ical advantages will translate into practice since the HMC method has a considerable overhead per iteration, nor it is clear in which way the better mixing truly pays-off. We perform a computational experiment in order to highlight the strengths and shortcomings of the proposed sampler against the simple but potentially effective Hoff algorithm. The desirable statistical properties of the extended rank likelihood are discussed elsewhere in detail in the references given, and as such we will focus solely on the computational aspects of inference.

## 3.1  Setup

We generate synthetic data from a 10-dimensional multivariate binary distribution with a full, randomly sample, copula correlation matrix. A dataset of 10,000 samples is used as a detailed case study in the next section and provides a typical scenario. Posterior inference is done using a inverse Wishart prior with 12 degrees of freedom and a identity matrix scaled by 12, as in [2], and defining the implied prior over the copula correlation matrix by standardization of matrices given by this prior. We focus on binary models because this allows us to evaluate the effect of a single boundary and how just this one barrier affects Hoff's algorithm. We focus on full correlation matrices instead of sparse inverse matrices [1] or low-rank decompositions [3] because this allows us a minimal interference of the mixing properties of other parameters and latent variables on our evaluation.

## 3.2  Illustrative Result and Analysis

Our HMC implementation is written in unoptimized MATLAB code, which is believed to severely slow down the procedure due to long loops of bouncing steps that could be much better handled in a compiled language. We note that in our implementation, the cost of each HMC step was roughly 50 times the cost of a Hoff step.

To achieve a reasonable running time we reduced our travel time to the relatively small $\pi/100$. Figure 2 summarizes the main findings with some choices of copula correlation coefficients out of the 10 dimensional case. The Supplementary Material has the full matrix. Color-coded depictions of copula correlations are shown, with sample-to-sample variability being relatively small for the HMC case after 30 iterations.

It is clear that the need for reducing travel time will not in general guarantee a high effective sample size. We observed increases of effective sample size at the order of a 1.5 to 5-fold improvement compared to the procedure by Hoff, which under our current MATLAB implementation does not seem to to justify the overhead. However, even at this modest level of travel time, the procedure does move the whole mass of underlying truncated Gaussian data points to its mode in far fewer iterations, when compared to Hoff's algorithm. This well within the expected behaviour, where the pure Gibbs procedure requires a large number of moves towards the right direction, since the whole data has to travel together based on changes at a single level at a time. Within this position, however, local exploration is somewhat similar for both methods. Our take-home message: with a sample size at the order of tens of thousands many bounces will be necessary, but even in this case there are good uses of an expensive HMC method that can bring a sampler efficiently to a proper location. While the plain full correlation model might have other simple ways of being initialized, this might not be clear with other extended rank likelihood models [3]. We provide a black-box approach that is by construction less sensitive to starting points, and has a built-in mechanism for trading-off the computational cost of its step against the quality of its mixing.