[Reviews · NeurIPS 2013]

Submitted by Assigned_Reviewer_4

The focus is on efficient posterior sampling in copula models for discrete data relying on the Hoff extended rank likelihood. Efficient posterior inference in copula models for discrete data is a topic of substantial interest in my view. The authors algorithm builds on the Hamiltonian MC approach of Pakman and Paninski (2012) for sampling from multivariate Gaussians subject to linear inequality constraints. However, they do not just directly apply the algorithm, but propose a novel approach to reduce the computational burden to O(n) in the sample size by relying on Hough Envelope algorithm. I find the paper very well written and to contain interesting ideas. Hence, the quality, clarity and originality is high.

However, I have questions about the significance (see below), which the authors have effectively addressed in their rebuttal. It is important for them to carefully revise their paper to include these improved experiments.

Previous concerns:
The experiment is obviously not compelling. I have applied the Hoff algorithm they are attempting to improve upon and haven't noticed major mixing problems, and Murray et al. (2013) additionally note good mixing in their copula factor model case relying on a similar sampling algorithm. The proposed algorithm is substantially more complicated and in the author's code has 50 times the computational burden per MCMC step while improving effective sample size only modestly. Examining the trace plots presented, it seems that perhaps convergence is substantially faster for the proposed algorithm but not mixing. Also, I wonder if the example shown is carefully chosen to show problems with Hoff's approach in involving a quite big sample size relative to the dimension of the model. In more modest sample sizes, I wonder whether such "gains" will be shown. Obviously it would be *much* more compelling to show gains in effective sample size per CPU time, while containing some discussion about the settings in which the gains are expected to be relatively small or large. It seems that Hoff wins easily under such a metric and additionally has the advantage of being substantially simpler. I would have trouble implementing the proposed algorithm based on the brief description given, but wouldn't bother anyway based on these results.

Summary: Interesting algorithm addressing an important problem and the paper is well written. In the initial submitted paper the experimental results were extremely weak. However, in the rebuttal, the authors included some additional much more compelling experiments. If these new experiments can be included, then I think the paper is acceptable.

Submitted by Assigned_Reviewer_5

Summary:
The authors develop a novel algorithm for Bayesian inference in copula models for multivariate discrete distributions. Their work follows and extends the work of Hoff (2007), who assumes a Gaussian copula. The aim is inference on the correlation matrix of the Gaussian copula, and Hoff (2007) proposes a Gibbs sampling algorithm that proceeds by first introducing latent variables Z. Conditioned on the latent variables Z and the data, the correlation matrix has an inverse-Wishart distribution and can easily be sampled. Conditioned on the data and the correlation matrix, the latent variables Z have a multivariate truncated normal distribution with regions defined by the order statistics of the data. This last step is the key contribution of the authors; whereas, Hoff (2007) used a separate Gibbs step for each latent Z_{i,j}, the authors proposing using recent advances in constrained Hamiltonian MCMC, which allows them to sample the latent (Z_{i,j}, i=1,...,n) jointly. This improves mixing and speeds up computations.

Strengths:
This is a nice, well written paper, and I particularly like how the authors highlight their contribution over previous work.
Although there are no methodological developments, the algorithm developed is practically useful, as it speeds up computations and improves mixing.

Weaknesses:
The authors do not mention the work of Damian and Walker (2001) who discuss techniques to sample from multivariate truncated normals using latent variables. How would this approach (or an extension) compare?
pg. 2, line 107, the authors could improve notation by possibly replacing the truncation region D by D(y), so that it is more explicit that the truncation region depends on the data. Otherwise, it appears that the posterior of Z and C doesn't depend on Y.
pg. 3 line 144, should this read log(p(x)) \propto … ?
pg. 5 line 258, the notation used in this sentence is not explained. Also should the sentence read Z_{i,j} conditioned on Z_{/i,/j} for Hoff's algorithm?

Quality: This is a technically sound paper that combines recent advances in Hamiltonian MCMC to improve Hoff's algorithm for Gaussian copula models of multivariate discrete distributions.

Clarity: This is a clear, well written paper. I especially appreciate the clear emphasis on the contribution of their work over previous work.

Originality: While there are no novel developments in this paper, they do apply Hamiltonian MCMC methods to produce a practically useful improvement over Hoff's algorithm.

Significance: The results are important, and I believe will likely be incorporated in extensions of the Gaussian copula models of multivariate discrete distributions.
Summary: The authors develop a novel algorithm for Bayesian inference in copula models for multivariate discrete distributions that improves upon the algorithm developed by Hoff(2002) by incorporating Hamiltonian MCMC methods. This results in improved mixing and faster computations.

Submitted by Assigned_Reviewer_6

This paper is concerned with learning Gaussian copulas after the empirical cdf is used to transform the marginals of all data. The challenge in this scenario seems to be linear constraints on the sampling domain, the number of which increases quadratically with the number of data points. The paper proposes a Hamiltonian Monte Carlo sampler where the constraints are handled by the sample trajectory "bouncing" off them. Empirical results are given, in particular showing superior performance, for the same number of steps, against the Hoff algorithm.

The paper seems technically sound, although I have some concerns regarding the empirical results. The presentation is fine, although note minor points below. I have not seen an approach such as this before and believe it to be original, although am not well-abreast of the copula literature, and my confidence score reflects this. The approach does seem quite computationally expensive, and the authors even admit that their MATLAB implementation is not particularly fast, which leaves question marks over potential impact.

My concerns with the empirical results are:

* The HMC is only run for 1000 steps, while the Hoff is run for 2000. The Hoff is clearly seen to make a few switches after the 1000 step mark, and a reader might worry that the HMC method could do the same. I presume this has been done because the HMC is much more expensive to run (the authors say 50 times so in Footnote 7). I don't doubt the final result, especially looking at the plots in the supplementary material, but for thoroughness it would be good to see the HMC run for 2000 steps, or a plot of posterior densities of each sample that indicates that the jumps of the Hoff algorithm are not due to a second mode, or a discussion point affirming that the posterior is definitely unimodal (which I think is knowable in this case a priori).

* I'd like to see more discussion on the relative computational expense of the HMC and Hoff methods. Footnote 7 makes an excuse, whereas I'd rather see an explanation. Should I expect a good implementation of HMC to be as fast as Hoff? At the moment, with HMC taking 50 times longer, it may be fairer to compare HMC at iteration #50 with Hoff at iteration #2500 in the bottom row of Figure 3.

A few minor points on presentation:

* There's a citation, [12], given for the first time in the conclusion. If this is important it should probably be introduced as related work in the introduction.

* The footnoting might be considered excessive. I'd suggest that some of these could be moved into the text without distracting the reader, especially 2, as it ends a paragraph anyway, but 3, 5, and 7 as well. In fact 5 should be, as it introduces notation, which is required knowledge rather than an aside, and I think 7 also, as, in the context of comments on the empirical results above, it's important to know that the HMC steps take a lot longer to run.
Summary: A decent paper, but a few question marks over the fairness of the empirical results.
Author Feedback

Author rebuttal: We thank our reviewers for their constructive criticism and commentary on improving our manuscript, and for the unanimous appreciation of the methodology per se, if not for the experiments. We agree that the experimental section should be overhauled to demonstrate the state-of-the-art impact of our contribution beyond any doubt.

For our reply we briefly describe new experiments that highlight the benefits of our HMC treatment, compared to a state-of-the-art parameter expansion (PX) sampling scheme.


INTRO:

In retrospect, it appears that for a typical reader our original experimental setup was too conservative to demonstrate the added benefits of HMC. In these new experiments we ask the important question

"How do the two schemes compare when we exploit the full-advantage of the HMC machinery to jointly sample parameters and the augmented data Z, in a model of latent variables and structured correlations?".

By the end of our rebuttal, we hope to convey that under such circumstances the superior convergence speed and mixing of HMC undeniably compensate for its computational overhead.


NEW EXPERIMENTAL SETUP:

Our application is to the Gaussian copula latent factor model of [13] (~ Hoff's model for low-rank structured correlation matrices). We modify the parameter expansion (PX) algorithm used by [13] by replacing two of its Gibbs steps with a single HMC step. We show a much faster convergence to the true mode with considerable support on its vicinity. We show that unlike the HMC, Murray et al.'s PX algorithm falls short of properly exploring the posterior in any reasonable finite amount of time, even for small models, even for small samples. Worse, *PX fails in ways one cannot easily detect*.

Namely, we sample each row of the factor loadings matrix Lambda jointly with the corresponding column of the augmented data matrix Z, conditioning on the higher-order latent factors. This step is analogous to Pakman and Paninski's [16,sec.3.1] use of HMC in the context of a binary probit model (the extension to many levels in the discrete marginal is straightforward). The sampling of the higher level latent factors remains identical to [13]. Our scheme involves no parameter expansion. We do however interweave the Gibbs step for the Z matrix similarly to Hoff. This has the added benefit of exploring the Z sample space within their current boundaries, complementing
the joint (lambda,z) sampling which moves the boundaries jointly. The value of such "interweaving" schemes has been addressed by Yu and Meng (JCGS, 2011).


NEW RESULTS:

Table of simulation setups we performed for both [13] (PX) and our method (HMC):

p (OBSERVED VARIABLES) k (LATENT FACTORS) M (ORDINAL LEVELS)
20 5 2
10 3 2
10 3 5

All setups involve 1000 observations in the data matrix Y and both schemes are run for 10000 iterations. Experiments were run on Intel COREi7 desktops with 4 cores and 8GB of RAM. Both methods are parallelized across the observed variables.

We produced two new plots for each setup:

1) Iteration-vs-RMSE plot:
Many functionals of the loadings matrix Lambda can be assessed. We focus on reconstructing the true (low-rank) correlation matrix of the Gaussian copula. In particular, we summarize the algorithm's outcome with the ROOT MEAN SQUARE ERROR (RMSE) of the differences between entries of the ground-truth correlation matrix and the implied correlation matrix at each iteration of a mcmc scheme (so the plot looks like a time-series of 10000 timepoints).
2) A histogram of the above plot summarizing the RMSE distribution.

Across setups, all Iteration-vs-RMSE plots share the following common characteristics:
- HMC and PX are identically initialized so their RMSEs rapidly drop from ~0.6 to ~0.4 within the first 5 iterations.

- Illustrative summary of the RMSE evolution:
p k M Method(elapsed time in seconds)
-----------------------------------------
20 5 2 HMC(6940s) RMSE=0.15 at iter #10, 0.11 at #30, then varies with a mode between 0.08 and 0.13 and a thick tail from 0.13 diminishing at 0.25, onwards until #10000. 30% of the posterior samples correspond to RMSE < 0.11.

20 5 2 PX (470s) RMSE=.4 at iter #10, 0.35 at #50, linearly drops to 0.13 at #500 then varies with a mode between 0.125 and 0.13, onwards until #10000. A negligible fraction of the PX posterior samples reach RMSE below 0.11.

MAIN MESSAGE: HMC reaches a better mode faster (iterations/CPUtime). Even more importantly the RMSE posterior samples of PX are concentrated in a much smaller region compared to HMC, even after 10000 iterations. This illustrates that PX poorly explores the true distribution. As an analogous situation we refer to the top and bottom panels of Figure 14 of Radford Neal’s Slice Sampling paper. If there was no comparison against HMC, there would be no evidence from the PX plot alone that the algorithm is performing poorly. This mirrors Radford Neal's statement opening Section 8 of his paper “a wrong answer is obtained without any obvious indication that something is amiss". The concentration on the posterior mode of PX in these simulations is misleading of the truth. PX might seen a bit simpler to implement, but it seems one cannot avoid using complex algorithms for complex models. We urge practitioners, including Reviewer_4, to revisit their past work with this model to find out by how much credible intervals of functionals of interest have been overconfident. Whether trivially or severely, the only game in town to check it out (we're aware of) is our own algorithm.

We have results analogous to the above for the rest of the setups (see table).